# Genome-Wide Association Studies of Biluochun Tea Plant Populations in Dongting Mountain and Comprehensive Identification of Candidate Genes Associated with Core Agronomic Traits by Four Analysis Models

**DOI:** 10.3390/plants12213719

**Published:** 2023-10-30

**Authors:** Xiaogang Lei, Haoyu Li, Pingping Li, Huan Zhang, Zhaolan Han, Bin Yang, Yu Duan, Ndombi Salome Njeri, Daqiang Yang, Junhua Zheng, Yuanchun Ma, Xujun Zhu, Wanping Fang

**Affiliations:** 1College of Horticulture, Nanjing Agricultural University, Nanjing 210095, China; 2020104081@stu.njau.edu.cn (X.L.); 2022104080@stu.njau.edu.cn (P.L.); 2021104080@stu.njau.edu.cn (H.Z.); 2021204035@stu.njau.edu.cn (Z.H.); 2020204037@stu.njau.edu.cn (B.Y.); t2022069@njau.edu.cn (Y.D.); sn.njeri1@gmail.com (N.S.N.); myc@njau.edu.cn (Y.M.); zhuxujun@njau.edu.cn (X.Z.); 2Dongshan Agriculture and Forestry Service Station, Suzhou 215100, China; 17761898233@163.com (H.L.); kongxinquan41@163.com (D.Y.); 15895425278@163.com (J.Z.)

**Keywords:** Biluochun tea plant, genome-wide association study, agronomic traits, single nucleotide polymorphisms, marker-assisted breeding

## Abstract

The elite germplasm resources are key to the beautiful appearance and pleasant flavor of Biluochun tea. We collected and measured the agronomic traits of 95 tea plants to reveal the trait diversity and breeding value of Biluochun tea plant populations. The results revealed that the agronomic traits of Biluochun tea plant populations were diverse and had high breeding value. Additionally, we resequenced these tea plant populations to reveal genetic diversity, population structure, and selection pressure. The Biluochun tea plant populations contained two groups and were least affected by natural selection based on the results of population structure and selection pressure. More importantly, four non-synonymous single nucleotide polymorphisms (nsSNPs) and candidate genes associated with (−)-gallocatechin gallate (GCG), (−)-gallocatechin (GC), and caffeine (CAF) were detected using at least two GWAS models. The results will promote the development and application of molecular markers and the utilization of elite germplasm from Biluochun populations.

## 1. Introduction

Tea plant [*Camellia Sinensis* (L.) O. Kuntze] is an important, woody, perennial, cash crop that belongs to the family Theaceae, Camellia genus. Tea, a non-alcoholic beverage, processed from fresh leaves of tea plant has a large medicinal and economic value. The tea industry has greatly promoted the development of China’s agricultural economy, especially in Guizhou, Yunnan, and Sichuan provinces. Tea is one of the most popular beverages in the world, enjoyed by more than 3 billion people in 160 countries. Large numbers of tea consumers are mainly attracted by its health benefits and pleasant flavor. Tea is rich in characteristic metabolites, such as polyphenols, caffeine, amino acids, and tea polysaccharides, which have been proven to have antibacterial, anti-cancer, anti-inflammatory, and anti-oxidant functions that lower blood pressure among other beneficial effects [1,2,3,4,5]. The unique flavor is the most impressive quality in tea. The unique flavor of tea is also formed by characteristic metabolites. Primary and secondary metabolites could form sour, sweet, bitter, and astringent tastes. The difference in tea flavor is formed by the difference in tea metabolites, which mainly contains catechins, amino acids, soluble sugars, purine alkaloids caffeine, terpene volatiles, and triterpene saponins [6]. Catechins contain (+)-catechin (C), (−)-epicatechin (EC), (−)-epigallocatechin (EGC), (−)-gallocatechin (GC), (−)-catechin gallate (CG), (−)-epicatechin gallate (ECG), (−)-gallocatechin gallate (GCG), and (−)-epigallocatechin gallate (EGCG). Each of them brings a unique flavor to the tea. Catechins give tea bitterness, amino acids provide umami and sweetness, soluble sugar gives sweetness, caffeine gives tea bitterness, tea saponin gives tea infusion bitterness, and alcohols, phenols, aldehydes, and esters form the attractive aroma of tea [7,8,9,10]. Catechins, caffeine, and L-theanine are very important metabolic components in tea; not only do they give the tea plant its unique flavor, they also have health functions [11,12]. The synthesis and regulation mechanism of these beneficial metabolites in tea need to be further studied. Therefore, we need to collect more tea plant resources combined with more advanced technologies to reveal the molecular mechanism of tea quality formation.

Whole-genome resequencing is a standard method to study plant origin, as well as evolution and genetic variations based on high-throughput sequencing technology, which can detect a large number of DNA markers, such as SNPs, indels, and copy number variations. With the publication of plant reference genomes, genome resequencing has begun to explore the genetic diversity of plant populations [13]. The greatest advantage of resequencing is the ability to map genetic variation at the whole-genome level, and the construction of genetic maps at the whole-genome level in many plants has facilitated marker-assisted selection and genomic selection methods to accelerate crop improvement [13,14]. More importantly, combining genome resequencing with plant phenotypic data such as yield, plant height, fruit size, fruit quality, and metabolite abundance facilitates the detection of loci for these traits [15,16,17]. Additionally, plant genome resequencing has greatly enhanced our understanding of crop domestication, plant adaptation, and evolution, and has identified loci that confer valuable traits [18,19,20]. The upgrading of sequencing technology and the reduction in sequencing cost will further promote the development of whole-genome resequencing. After the publication of the chromosome-level genome of tea plant, the whole-genome resequencing of tea plant has been greatly promoted [21]. Based on whole-genome resequencing data, the genetic diversity, population structure, origin, and evolution of tea plants have been revealed, and candidate genes regulating tree type, leaf color, spring bud flush of tea, and characteristic metabolites have been identified [22,23,24,25,26,27,28,29]. The high heterozygosity in the tea plant genome hinders the identification of key genes that regulate the phenotype and characteristic metabolites of the tea plant.

Biluochun tea is one of the top ten traditional famous teas in China, which originated in Dongting mountain, Wuzhong district, Suzhou city, Jiangsu province, China. Biluochun tea is famous for its beautiful appearance, sweet and fresh taste, and floral and fruity fragrance. Fresh tea leaves from Dongting mountain are the best raw materials for processing Biluochun tea, which is the key to its superb quality. The diverse phenotypes and abundant characteristic metabolites of Biluochun tea plant populations form the excellent appearance, taste, and fragrance of Biluochun tea. Therefore, the population structure, evolution mechanism, phenotype diversity, and quality-related metabolites variation of Biluochun tea plant populations need to be elucidated. In this study, we resequenced Biluochun tea plant populations (over 10×) from Dongting mountain. The phylogenetic relationships, population structure, and linkage disequilibrium (LD) pattern of tea plant populations were evaluated, and the molecular regulation mechanism of tea plant phenotype and characteristic metabolites were studied based on genome-wide association analysis. Expression patterns of genes identified by GWAS in different haplotypes were validated by qRT-PCR. Our findings will provide valuable insights for breeding the specific cultivar of Biluochun tea and the development of molecular markers in the future.

## 2. Result

### 2.1. Metabolic Variation

Fourteen flavor-related compounds including (+)-catechin (C), (−)-epicatechin (EC), (−)-epigallocatechin (EGC), (−)-gallocatechin (GC), gallic acid (GA), (−)-catechin gallate (CG), (−)-epicatechin gallate (ECG), (−)-gallocatechin gallate (GCG), (−)-epigallocatechin gallate (EGCG), caffeine (CAF), total catechins (TC), total polyphenols (TP), soluble sugar (SS), free amino acids (FAA), and growth-related chlorophyll (Chl) were used to detect the distribution and variation of metabolites among 95 tea plant varieties. The results showed that the content of various metabolites in 95 tea plant varieties varied greatly with high variability (Appendix A). The content of CAF had a great variation ranging from 1.65% to 7.09% with a coefficient of variation (CV) of 44.27%, whereas C showed the smallest range from 0.87% to 1.04% with a CV of 4.15%. Except for C, the CV of all the other characteristic metabolites in the 95 tea plant varieties exceeded 10%, indicating high diversity. Therefore, Biluochun tea plant populations may have a large breeding value. Additionally, the Shannon–Wiener index values of all metabolites were between 6.15 and 6.53 with less difference, demonstrating that these tea plants contained relatively high richness. More importantly, the frequency distributions of most traits showed approximately normal distributions (Figure 1 and Appendix A). Therefore, these tea plants can be used as ideal materials to study the metabolic mechanism of tea characteristic metabolites.

Furthermore, the Pearson correlation coefficient (PCC) between these five normally distributed and important characteristic metabolites (SS, FAA, TP, TC, and CAF) was estimated (Figure 1). Statistically significant differences were shown between every pair of metabolites with *p*-value < 0.01. There was a strong correlation between most of these metabolites. However, the correlations between FAA and three metabolites (expect for CAF) were weak (|PCC| ≤ 0.40). There was a significant negative correlation between FAA and CAF (PCC = −0.70), indicating that tea infusions with strong umami tend to be less bitter. Then, a significant negative correlation of SS with TP, TC, and CAF was observed (PCC ≥ −0.70), indicating that bitter and astringent tea infusions tend to be less sweet. Interestingly, CAF was positively correlated with TP and TC (PCC = 0.70), suggesting a positive correlation between bitterness and astringency of tea infusion (Figure 1). Additionally, the PCC between all characteristic metabolites was processed (Appendix A). Except for EGC and EC, the correlation between most characteristic metabolites was strong. Interestingly, SS was significantly negatively correlated with almost all metabolites, while there is a strong positive correlation between TP, TC, CAF and most ester catechins. Catechins and polyphenols affect tea taste, and contribute to the astringency of the tea. They are also precursors of theaflavins, which are golden yellow pigments and contribute to tea soup color. Caffeine is also an important factor affecting the quality of tea. It is strongly water soluble and contributes to the bitter taste of the tea. These results showed that high levels of soluble sugar in tea infusion may reduce tea quality and there is a strong correlation between bitterness and astringency in tea soup. All the results indicated that there was a close relationship among these characteristic metabolites, which played an important role in the formation of good quality of tea (Appendix A).

### 2.2. Phenotypic Variation

To reveal the diversity of phenotypic traits, the leaf length, width, and area of 95 tea plant varieties were measured. The results showed that the phenotypic traits of Biluochun tea plant populations had high diversity and richness, similar to the distribution of metabolites (Appendix A). For instance, the leaf length varied greatly, from 23.17 to 54.69 mm, with a mean of 37.27 mm. Similar to leaf length, the leaf area ranged from 200.55 to 878.11 mm^2^. The CV values of leaf length, width, and area were 15.90%, 13.24%, and 29.67%, respectively. More importantly, the frequency distributions of all three phenotypic traits approximate a normal distribution (Appendix A). The leaf phenotypes of Biluochun tea plant populations vary greatly; therefore, they can be used as important germplasms for studying the molecular regulation mechanism of leaf shape.

### 2.3. Sequencing and Variant Discovery

A total of 95 Biluochun tea plant varieties from Dongting mountain were collected. According to the place of origin, these tea plants were divided into 6 subgroups, including 61 MLS, 8 BLL, 7 BLX, 9 ZW, 8 SJ, and 2 BW subgroups.

The Illumina HiSeq 10× sequencing platform was used to resequence 95 Biluochun tea plant varieties. As a result, a total of 19.332 billion raw 150 bp paired-end reads were generated, and 2.83 Tb of clean data remained with an average coverage depth of more than 10×. Our resequencing reads were mapped onto the published *Camellia sinensis* var. *sinensis* ‘Shuchazao’ genome. A total of 51,336,279 SNPs and 2,964,483 indels (insertions and deletions, range 1–67 bp, mean 4 bp) were identified. Of the 19,624,326 filtered SNPs (allele number larger ≥ 2, MAF < 0.05, and MAC ≤ 3), 2,179,998 and 323,284 SNPs were distributed in non-coding and coding sequences, respectively. Moreover, 171,894 non-synonymous SNPs (nsSNPs) were identified in 29,095 genes, and 6799 frameshift indels were identified in 4791 genes (Figure 2a,b). In order to evaluate the selective effect of environment on tea plant growth, the ratio of nsSNPs to synonymous SNPs (dN/dS) was calculated, and the result was 1.17. Additionally, among all identified SNPs, 1.6% were located in coding regions: 0.88% were non-synonymous and 0.75% were synonymous (Appendix A). The results showed that the proportion of nsSNPs in the coding regions of Biluochun tea plant populations was significantly low, indicating that the genetic variation in the coding region of Biluochun tea plant populations was less.

### 2.4. Phylogenetic Analysis and Population Structure of Biluochun Tea Plant Populations

A total of 95 Biluochun tea plant populations of the same origin region were clustered into two groups, according to the population structure analysis (Appendix A). In order to further analyze the population structure of Biluochun tea plant populations, the principal component analysis (PCA) was performed based on 19,624,326 filtered SNPs. Two groups were distinguished by PC1, PC2, and PC3, respectively (Figure 3a,b). Then, neighbor-joining (NJ) phylogenetic trees were constructed based on the genetic distances derived from SNP differences in these individuals (Figure 3c). Biluochun tea plant populations can be divided into two distinct groups, group I (BL I) and group II (BL II). BL I contained 46 tea plant varieties, most of which belonged to MLS subgroup, and only 4 tea plant varieties belonged to ZW subgroup. BL II contained 49 tea plant varieties of all subgroups. According to the results of phylogenetic tree and population structure analysis, these tea plant varieties were also divided into two groups, which was the same as the result of PCA.

Then, the filtered SNPs were used to estimate the LD decay distance. The genome-wide LD decay rate for all populations is approximately 100 bp, with r^2^ dropping to half of its maximum (Figure 3d). Moreover, tea plants from ZW subgroup had the highest LD decay rate (r^2^ = 500 bp), and tea plants from MLS subgroup had the lowest LD decay rate (r^2^ = 200 bp). Moreover, one SNP per 9 kb is the theoretical average marker density. However, in this study, the actual value of the SNP density in the genome has reached 6.25 (SNP/kb), and the distribution of these SNPs in the whole genome is shown in Appendix A. Since the SNP density of Biluochun tea plant population is much higher than the theoretical average marker density, it is necessary to identify SNPs related to the agronomic traits of Biluochun tea plant.

### 2.5. Genome-Wide Association Studies

All three phenotypic traits and 14 metabolites (except for GA) were integrated with whole-genome resequencing data, and a large number of SNPs was identified. Then, four GWAS models (GLM, GLM(Q), MLM(K), MLM(QK)) were used to identify candidate SNPs (Figure 4 and Appendix A). For instance, based on the GLM model, there were 3, 4, 26, 14, 30, 151, 219, and 0 SNPs corresponding to 4, 3, 22, 9, 15, 261, 317, and 0 candidate genes, respectively. These genes (*p*-value < 3.89 × 10^−9^) were found to be closely related to SS, FAA, TP, TC, CAF, GC, GCG, and Chl. Additionally, in order to obtain reliable SNPs and genes, quantile–quantile (QQ) plots were screened. QQ plots for CAF, GC, GCG, SS, FAA, TP, TC, and Chl are approximately the optimal curve, indicating that the false positive error is low and the results of these models are reliable (Figure 4a and Appendix A). Therefore, the above eight metabolites all met the basic requirements of GWAS and can be used to identify candidate SNPs related to the metabolites.

A total of 478 SNPs (*p*-value < 3.89 × 10^−9^) that were significantly associated with the above seven metabolites were simultaneously identified by at least two GWAS models (Appendix A). Among these, 3, 5, 23, 14, 30, 178, and 223 SNPs were found to be correlated with SS, FAA, TP, TC, CAF, GC, and GCG, respectively. All SNPs for GC and GCG were identified and they were widely located on all 15 chromosomes. For SS, three candidate SNPs were located on chromosomes 1, 3, and 9. For FAA, five candidate SNPs were located on chromosomes 1, 2, and 3. For TP, 23 candidate SNPs were located on seven chromosomes, namely, chromosomes 1, 2, 3, 6, 9, 11, 12, and 14. For TC, 14 candidate SNPs were located on five chromosomes, namely, chromosomes 1, 3, 5, 9, and 12. A total of 30 SNPs that were significantly associated with CAF were identified and located on chromosomes 1, 3, 5, 6, 7, 9, 11, 12, 14, and 15. Of these, 114 candidate SNPs were found simultaneously by at least three GWAS models. A total of 109 candidate SNPs were simultaneously found by four GWAS models. These candidate SNPs were used to identify the candidate genes that regulate metabolites.

### 2.6. Prediction of Potential Candidate Genes

Tea leaves contain rich secondary metabolites, which are crucial to the formation of good quality tea. Presently, amino acids, catechins, caffeine, tea polyphenols, and soluble sugars are the most studied quality-related metabolites. Candidate SNP loci were identified simultaneously in more than two GWAS models. Here, we identified a total of four nsSNPs which are closely related to GC, GCG, and CAF, and thus considered as candidate SNPs (Appendix A). One nsSNP (qCAF-9-5) was identified to be significantly associated with CAF, located in *CSS0039426* (*p*-value = 2.38 × 10^−9^). The result of functional annotation showed that *CSS0039426* was involved in encoding CREB-regulated transcription coactivator 2 (Table 1). One nsSNP (qGC-9-14) significantly associated with GC was located at *CSS0044862* (*p*-value = 4.57 × 10^−10^), and this gene is involved in the signal transduction process (Table 1). Additionally, two nsSNPs (qGCG-9-40 and qGCG-9-23) were significantly associated with GCG, and these two nsSNPs were located at *CSS0026218* and *CSS0027172* (*p*-value = 5.51 × 10^−10^ and 1.75 × 10^−9^), respectively. These results of functional annotations showed that these two genes are predicted as LysM-containing receptor-like kinase and tubby-like protein, respectively (Table 1).

One nsSNP (qCAF-9-2) was significantly associated with CAF, and caused the 176th base mutation from C to G in the exon region of *CSS0039426*. Accordingly, the amino acid sequence changed from Ala to Gly at residue 56. One nsSNP (qGC-9-14) was significantly associated with GC, and the 209th base changed from A to G in the exon region of *CSS0044862*. Accordingly, the amino acid sequence changed from Tyr to Cys at residue 73. Additionally, two nsSNPs (qGCG-9-40 and qGCG-9-23) were significantly associated with GCG, which caused the 629th base mutation from A to G in the exon region of *CSS0026218* and the 886th base mutation from A to U in the exon region of *CSS0027172*, correspondingly, the amino acid sequence of CSS0026218 changed from Glu to Gly at residue 210, and the amino acid sequence of *CSS0027172* changed from Thr to Ser at residue 296.

Based on the missense mutations in four genes, we defined four pairs of haplotypes of *CSS0039426* (HapA and HapB), *CSS0044862* (HapC and HapD), *CSS0026218* (HapE and HapF), and *CSS0027172* (HapG and HapH). Then, we isolated and aligned homologous sequences of *CSS0039426*, *CSS0044862*, *CSS0026218*, and *CSS0027172* from different tea plant haplotypes (Appendix A). Additionally, we compared the content distribution of the three metabolites in different haplotypes of tea plants. These tea plant varieties with the favorable HapB and HapD displayed significantly higher GCG content than those with the HapA and HapC types, respectively (Appendix A). These individuals with the favorable HapF displayed significantly higher GC content than those with the HapE types (Appendix A). These varieties with the favorable HapH displayed significantly higher CAF content than those with the HapG types (Appendix A). These results indicated that the content of GCG, GC, and CAF was significantly increased in tea plant varieties with favorable haplotype variants compared with those varieties from other haplotypes.

In order to explore the expression patterns of candidate genes among different haplotypes, the expression levels of four representative genes identified by GWAS in four pairs of haplotypes were studied by real-time quantitative PCR (RT-qPCR) (Figure 4b). The results showed that the expression levels of GCG-related genes (*CSS0026218* and *CSS0027172*) in high GCG tea plant varieties (HapB and HapD) were significantly higher than that in low GCG tea plant varieties (HapA and HapC). The expression levels of GC-related gene (*CSS0044862*) in high GC tea plant varieties (HapF) were significantly higher than that in low GC tea plant varieties (HapE). The expression levels of CAF-related gene (*CSS0039426*) in low caffeine tea plant varieties (HapG) were significantly lower than that in high caffeine tea plant varieties (HapH). These results indicated that the expression levels of the four genes and the contents of the corresponding metabolites had the same trend of change. Specifically, the higher the metabolites content, the higher the expression levels of the corresponding genes. Furthermore, these results suggest that the content of quality-related metabolites is controlled by multiple genes. These nsSNPs can be used as molecular markers to quickly predict the content of CAF, GC, and GCG in different tea plant varieties.

## 3. Discussion

Dongting mountain is the birthplace of Biluochun tea. Biluochun tea is famous for its good quality and beautiful appearance. The fresh tea leaves processed into Biluochun tea are collected from the Biluochun tea plant populations in Dongting mountain. However, there is no research about Biluochun tea plant populations. In this study, trait and genetic datasets of Biluochun tea plant populations were integrated to explore the population structure, genetic diversity, and evolutionary selection of Biluochun tea plant populations, as well as the genetic mechanisms based on metabolites and leaf phenotypes.

Cultivated tea plant populations have a wide variety of traits and genetic diversity [30]. Based on the phenotypic and metabolic data of 95 Biluochun tea plant varieties, the diversity of these traits was analyzed. The Shannon–Wiener index values of all traits were greater than 6 (Appendix A), indicating a high diversity of traits in these tea plant cultivars [30,31]. Except for C, GCG, and CAF, the coefficient of variation of other traits was greater than 10%. Tea plants are self-incompatible plants, and the heterozygosity of seeding tea plants is higher compared to clonal tea plants. Therefore, there is high diversity and differences in the traits of seeding tea plant varieties, and they may contain abundant excellent traits and genetic resources, which have high breeding value. Therefore, it is very necessary to collect, protect, and study these highly variable seeding tea plants. Based on these tea plant germplasms, we can study the molecular mechanism of tea plant trait regulation and develop corresponding molecular markers for rapid breeding.

To assess selection constraints of Biluochun tea plant populations in natural habitats, the ratio of nsSNPs to synonymous SNPs (dN/dS) was calculated. The results showed that the dN/dS of the Biluochun tea populations was 1.17, which was slightly higher than the dN/dS (1.05) of the ancient tea plant populations in Yunnan and Guizhou provinces, indicating that the influence of natural selection on the Biluochun tea plant populations was slightly higher than that of the ancient tea plant populations in Yunnan and Guizhou provinces. However, the influence of natural selection on Biluochun tea plant population is still low [27]. The distribution of all SNPs showed that 0.88% of nsSNPs and 0.75% of synonymous SNPs were located in coding regions. The results showed that the proportion of nsSNPs in the coding region of Biluochun tea plant varieties was significantly lower than that of fruit trees (>5%), rice (1.64%), and ancient tea plants in Guizhou and Yunnan (1.12%) provinces. It shows that the genetic variation in the coding region of Biluochun tea plant varieties is less than that of fruit trees, annual crops, and ancient tea plants in Yunnan and Guizhou provinces [27,32,33,34].

### 3.1. Population Structure and Genetic Evolution Analysis

The origin and population structure of cultivated Biluochun tea plant varieties have not been revealed, thus hindering the research on Biluochun tea plants molecular breeding. The phylogenetic analysis of Biluochun tea plants showed that 95 tea plant varieties were divided into two groups, BL I and BL II. BL I contained four ZW and MLS, suggesting that gene exchange has occurred between them. There was also frequent gene exchange between the BL II groups including some members of all subgroups, which was caused by the continuous introgression of Biluochun tea populations during long-term cultivation [27]. Based on the phylogenetic tree results, a large number of tea plants belonging to the MLS subgroup was in the outermost branch of the phylogenetic tree and was older than the other five subgroups. Some tea plants belonging to the MLS subgroup were sister clades of the other five subgroups, suggesting that frequent gene exchange may exist between the tea plants of the MLS subgroup and those of other subgroups. Furthermore, members of the SJ, ZW, and BW subgroups clustered outside the clades of the BLL and BLX subgroups, suggesting that tea plant varieties from ZW, SJ, and BW subgroups were older than tea plant varieties from BLL and BLX subgroups. Additionally, the results of PCA and population structure analysis showed that 95 tea plant varieties were divided into two groups, some members of the MLS subgroup were divided into one group, and the rest of the tea plants were divided into the other group. The results of LD analysis showed that the LD of the six subgroups was less than 1 kb, and the LD of ZW was the largest (r^2^ = 500 bp), indicating that the selection pressure of these tea plants was low. Compared with Guizhou ancient tea plant (r^2^ = 19.3 kb) [27], Guizhou cultivated tea plant (r^2^ = 2 kb) [23] and wild tea plant (r^2^ = 116 bp) [35], indicating that the selection pressure of Biluochun tea plant population was lower and Biluochun tea plant varieties may have higher genetic diversity. Therefore, they are recommended as materials for studying the evolution, origin, and breeding of tea plants. 

### 3.2. Dissecting Four Candidate Genes of Quality-Related Traits

GLM and MLM are widely used to identify genetic variations associated with many tea plant phenotypes and flavor metabolites [23,24,29]. MLM and GLM are two classical GWAS models that are widely applied in the identification of candidate genes [36,37]. Although there are many efficient GWAS models, GLM and MLM are still widely applied in plant genetics and breeding as classic GWAS models [38,39,40,41,42,43]. Given that each GWAS model has different strengths and SNP identification methods, several GWAS models are simultaneously used to identify target SNPs for complex agronomic traits [44,45,46].

In this study, based on seven quality-related metabolites, 478 SNPs were simultaneously identified by more than two GWAS models. A total of 3, 5, 23, 14, 30, 178, and 223 SNPs were simultaneously identified by multiple GWAS models and correlated with SS, FAA, TP, TC, CAF, GC, and GCG content, respectively. Then, the SNPs were further identified, and four nsSNPs loci were reserved, which were located on four candidate genes (Table 1). Based on KEGG annotation and functional analysis of homologous genes in *Arabidopsis*, with reference to these genes, four candidate genes are annotated. These genes might be involved in regulating tea quality-related metabolites. In *Arabidopsis*, LysM-containing receptor-like kinase is involved in the immune response and stress, which can improve the ability of *Arabidopsis* to respond to stresses [47,48]. It is speculated that *CSS0026218* related to the synthesis of GCG may be involved in the immune and stress responses of plants, which is consistent with previous studies on the involvement of catechins in stress and immune responses of tea plants [49,50]. In *Arabidopsis*, tubby-like protein is involved in abiotic stress signal transduction dependent on ABA [51,52], which is consistent with the signal transduction pathway reported in tea plant that GC, GCG, and ABA are involved in abiotic or biological stress induction, suggesting that *CSS007172* may be involved in regulating this process [53]. Additionally, similar to *CSS0026218*, in *Arabidopsis*, a plant protein (DUF247) is associated with a QTL for quantitative disease resistance [54]. Therefore, it is speculated that *CSS0044862* associated with GC may participate in catechin-responsive plant immunity, which is consistent with the study of catechin-responsive plant immunity [53]. However, the homologous gene *CSS0039426* had not been identified in *Arabidopsis*, and KEGG annotation indicated that they were involved in encoding CREB-regulated transcription coactivator 2 (Table 1). Therefore, the function of these genes needs to be verified further by molecular experiments.

## 4. Materials and Methods

### 4.1. Plant Materials

A total of 95 tea plants were collected from the Biluochun tea germplasm repository (31°10′ N, 120°21′ E) in Dongting mountain, Wuzhong district, Jiangsu province, China. These tea plants showed great differences in phenotypic traits and quality-related metabolites, which could represent the morphological and genetic diversity characteristics of Biluochun tea plant populations.

### 4.2. Data Analysis

For each trait, we calculated the Shannon–Wiener diversity index (H′) as:(1)$$Shannon–Wiener\;index=-\sum P_{i}\ln P_{i}\;(i=1,2,3,4,\ldots ,n)$$
where *P_i_* is the value corresponding to each trait of tea plant sample, and n is the number of tea plant samples [55].

### 4.3. Sequencing, Mapping, and SNP Calling

Total genomic DNA was extracted from the samples and at least 3 ug genomic DNA was used to construct paired-end libraries with an insert size of 300–400 bp using the paired-end DNA Sample Prep Kit (Illumina Inc., San Diego, CA, USA). These libraries were sequenced using novaseq6000 (Illumina Inc., San Diego, CA, USA) NGS platform at Genedenovo company (Guangzhou, China). The reference genome sequence of the tea plant (*Camellia sinensis* var. *sinensis* cv. ‘Shuchazao’) was downloaded from the TPGDB database (http://eplant.njau.edu.cn/tea/, 16 September 2022) [56]. To identify SNPs and indels, the Burrows–Wheeler Aligner (BWA v 0.6.17) was used to align the clean reads from each sample against the reference genome with the default parameters [57]. Variant calling was performed for all samples using GATK’s Unified Genotyper (GATK v 3.4-46) [58]. SNPs and indels were filtered using GATK’s Variant Filtration with proper standards (−Window 4, −filter “QD < 2.0 || FS > 60.0 || MQ < 40.0”, −G_filter “GQ < 20”) [59]. Markers (SNPs/indels) with allele number larger than 2, missing data greater than 20%, minor allele frequency lower than 5% (0.01% for SAIGE and REGENIE), minor allele count (MAC) lower than 3 (for SAIGE and REGENIE only), and heterozygosity greater than 80% were removed using PLINK (v 1.9) software [60]. Then, a total of 19,958,095 SNPs and 1,338,594 indels remained. The remaining SNPs were used for population genetic analysis and all markers (including SNPs and indels) were used for GWAS.

### 4.4. Phylogenetic Tree and Population Genetics Analysis

We constructed a phylogenetic tree by a neighbor-joining method using the software PHYLIP (v 3.69) and the bootstrapped confidence interval was 1000 [61]. The population subdivision pattern was preliminary classified in the principal component analysis (PCA) by software GCTA (v 1.93.3) [62]. The admixture model-based software Admixture (v 1.3.0) [63] was further used to estimate the population structure (Q matrix). The tested K was set from 1 to 9, and the optimal K was determined with the lowest cross-validation error. The kinship matrix (K matrix) was calculated using the method of Loiselle et al. [64] by the software of SPAGeDi (v 1.5) [65]. The ancestry bar plots were plotted by R package pophelper (v 2.2.7) [66].

### 4.5. Linkage Disequilibrium Analysis

To evaluate the LD pattern among different tea plant populations, the squared allele frequency correlation (r^2^) between pairwise SNPs was computed by PopLDDecay (v 3.4.1) with default parameters [67]. The LD decay graphs were plotted using R package ggplot2 [68].

### 4.6. Genome-Wide Association Study

In our study, GWAS was performed in 95 tea plant varieties with 19,958,095 high-quality SNPs. A general linear model (GLM) was carried out for the single-locus method to evaluate the trait-SNP association analysis for tea plant phenotypes and metabolites using the Tassel (v 3.0.174) software [69]. The first three principal components (PCs) and kinship matrix were used as covariates to correct the population structure for decreasing false-positive rate in MLM. Bonferroni correction threshold (*p*-value = 0.01/marker number or 0.05/marker number) was used to identify significant associations [70]. The *p*-value of SNPs less than the Bonferroni correction threshold was selected as significant SNPs.

Genome-wide association mapping was implemented in GCTA (v 1.92.2) software [38], using the following models: (1) Simple linear model (LM or GLM). (2) General linear model only accounts for population structure as a fixed effect (GLM(Q)). (3) Mixed linear model only accounts for kinship as a random effect (MLM(K)). (4) Mixed linear model accounts for both population structure and kinship as fixed and random effects, respectively (MLM(Q + K)). All these four methods used the PCA and kinship matrices in our study. The Manhattan and quantile–quantile plots for GWAS were displayed using the R package CMplot [71].

### 4.7. Identification of Putative Genes

The significant SNPs identified by at least two different GWAS methods were regarded as the putative candidate loci. Local LD blocks containing at least two SNPs were calculated with all imputed SNPs using PLINK (v 1.9) software [72]. The local LD blocks of each significant SNP were determined via confidence intervals described by Gabriel [73]. The LD heatmap was visualized using Haploview (v 4.2) software [74]. All the genes located in the LD block of QTNs were extracted for further analysis.

### 4.8. Expression Pattern Analysis

All tea plant samples were frozen in liquid nitrogen and stored at –80 °C for the following steps. The RNA of samples was extracted using FastPure Plant Total RNA Isolation Kit (rich in polysaccharides and polyphenolics) from VAZYME (Vazyme Biotech Co., Ltd., Nanjing, China). The first-strand cDNA was synthesized using FastPure Viral DNA/RNA Mini Kit Pro (Vazyme Biotech Co., Ltd., Nanjing, China). The quantitative PCR primers were designed by Primer Premier (v 5) (Appendix A). Quantitative PCR was conducted in HiScript II One Step qRT-PCR SYBR Green Kit (Vazyme Biotech Co., Ltd., Dalian, China). The thermocycler was set as follows: 50 °C for 30 min; 95 °C for 30 s; 40 cycles of 95 °C for 10 s and 60 °C for 30 s; 95 °C for 30 s, 60 °C for 60 s, and 95 °C for 15 s. *CsACTIN* (CSS0008920, actin 7 isoform 1) were determined as references [75]. Quantitative expression analysis of each sample was carried out and relative gene expressions were analyzed using the 2^−∆∆Ct^ method [76]. All samples contained three biological and technical replicates.

### 4.9. Determination of Each Quality Index of Tea

#### 4.9.1. Chemicals

The standards for (+)-catechin (C), gallic acid (GA), (−)-epicatechin (EC), (−)-gallocatechin (GC), (−)-epigallocatechin (EGC), (−)-epicatechin gallate (ECG), (−)-catechin gallate (CG), (−)-epigallocatechin gallate (EGCG), (−)-gallocatechin gallate (GCG), and caffeine (CAF) were purchased from Yuanye (Shanghai yuanye Bio-Technology Co., Ltd., Shanghai, China). Anthrone was provided from Ourchem (Sinopharm Chemical Regent Co., Ltd., Shanghai, China). Acetonitrile and ethanol absolute were provided from GHTECH (Guangdong Guanghua Sci-Tech Co., Ltd., Guangdong, China). Methanol, HPLC grade, was obtained from TEDIA (Tedia Company, Inc., Fairfield, OH, USA). Formate, HPLC grade, was purchased from Aladdin (Aladdin Industrial Corporation, Shanghai, China). Deionized water was obtained from an in-house Milli-Q water purification system (Millipore, Bedford, MA, USA).

#### 4.9.2. Determination of Chlorophylls

The contents of chlorophyll *a*, *b* and carotenoids in the tea leaves were determined using three biological replicates. Pigments were extracted from 0.1 g young leaf samples using 10 mL of extraction solvent (ethanol) for 24 h in the dark. According to Porra’s method [77], chlorophyll *a*, *b* and carotenoids were quantified at wavelengths of 647, 664, and 470 nm using MRX II Dynex plate reader (Dynex Technologies Inc., Chantilly, VA, USA). 

#### 4.9.3. Determination of Free Amino Acids

The total amount of free amino acid were determined 2013 by ninhydrin colorimetry according to Wen (2020) with slight modifications [78]. Briefly, each individual tea sample (300 mg) was ground into a homogenous powder and extracted with 45 mL of distilled water heated at 100 °C for 45 min, then infusion was obtained by filtration. After it was cooled at room temperature, the infusion returned to a volume of up to 50 mL with distilled water. Following this, 1 mL of the obtained filtrate (as the infused tea sample), 0.5 mL buffer, and 0.5 mL ninhydrin reagent were heated at 100 °C for 45 min. After it was cooled at room temperature, the infusion returned to a volume of up to 25 mL with distilled water. The standard curve of theanine solution was used for quantitative analysis (Appendix A). The absorption value was recorded by an ultraviolet spectrophotometer at 570 nm. The injection volume was 200 µL.

#### 4.9.4. Determination of Soluble Sugar

The quantitative analysis of soluble sugar was based on previous methods, with some modifications. Specifically, for extraction, 0.1 g of tea powder sample was infused in 8 mL of boiling deionized water for 30 min in a boiling water bath. The other steps involved were identical to those in a previous study [79]. The standard curve of glucose solution was used for quantitative analysis (Appendix A). The absorption value was recorded by an ultraviolet spectrophotometer at 620 nm. The injection volume was 200 µL.

#### 4.9.5. Determination of Tea Polyphenols

Tea polyphenols content of tea leaves was measured according to Wen (2020) with slight modifications [78]. This method was slightly modified, and the specific steps were as follows. Briefly, the fresh tea leaves were placed in the microwave oven (M1-L201B, Midea Group Co., Ltd., Foshan, China) for 8 min for fixation, followed by drying at 90 °C to a constant weight. Dry tea leaves were ground into powder and uniformly sieved through an 80 mesh sieve to obtain tea powder. Then, 0.2 g of tea powder sample was infused in 5 mL of 70% methanol, which was added after heating in an electrically heated thermostatic water bath. The other steps involved were identical to those in a previous study [80]. The standard curve of gallic acid solution was used for quantitative analysis (Appendix A). The absorption value was recorded by an ultraviolet spectrophotometer at 765 nm.

#### 4.9.6. Determination of Catechins and Caffeine

The content of catechins and caffeine was determined according to Wen (2019) with slight modifications [81]. The content of catechins and caffeine was determined by UltiMate 3000 HPLC (Thermo Fisher Scientific Inc, Carlsbad, CA, USA). The specific steps involved were identical to those in a previous study [81]. The injection volume for all of the samples was 20 μL. The gradient mobile phase comprised a solution of A (5% acetonitrile + 0.26% phosphoric acid) and B phase (80% methanol solution). The other conditions included chromatographic conditions (detection wavelength 280 nm), flow rate (0.8 mL·min^−1^), column temperature (40 °C), injection volume (2 μL), and mobile phase gradient elution. The standard curve of catechins and caffeine solution was used for quantitative analysis (Appendix A). All the samples were determined in three replications.

#### 4.9.7. Determination of Morphological Parameters of Tea Leaf

The length, width, circumference, and area of tea leaves were measured by scanmarker i800plus (Microtek Co., Ltd., Shanghai, China) and each sample contained fifteen replicates. All values were stated as the mean ± SD for fifteen replications.

## 5. Conclusions

Based on plant phenotypes, physiology data, and whole-genome resequencing technology, this study reveals the population structure, evolutionary process, phylogenetic relationship, phenotypic diversity, and genetic diversity of the Biluochun tea plant population in Dongting mountain. At least two GWAS models were used to identify 478 significant SNPs associated with the seven quality-related metabolites. Among these SNPs, four nsSNPs (qCAF-9-5, qGC-9-14, qGCG-9-40, and qGCG-9-23) were closely related to the content of CAF, GC, and GCG. Furthermore, based on four nsSNPs, four key candidate genes (*CSS0039426*, *CSS0040558*, *CSS0026218*, and *CSS0027172*) were identified. In conclusion, four robust nsSNPs and candidate genes might be involved in the regulation of the synthesis of tea quality-related metabolites, which provided important resources for the selection of excellent tea plant varieties and the development of molecular markers.

## Figures and Tables

**Figure 1 plants-12-03719-f001:**
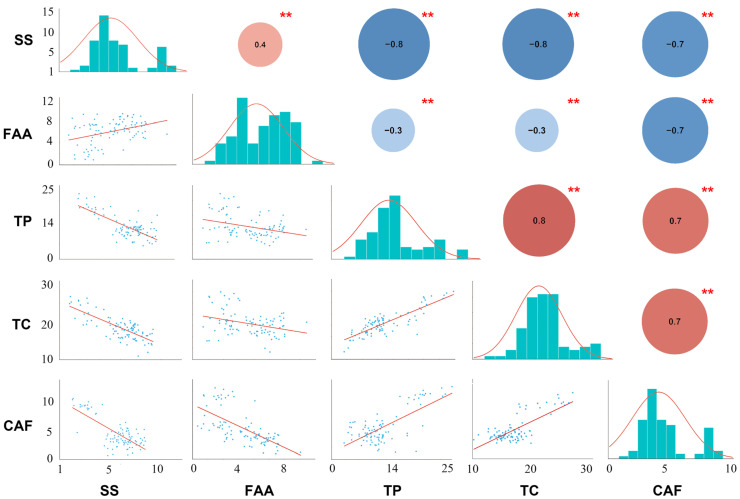
Distribution of five quality-related metabolites in tea plant and Pearson coefficient analysis. The lower left represents the linear regression statistics between each component, the diagonal histogram represents the distribution of each trait, and the upper right number represents the correlation coefficient (positive numbers and red represent positive correlation, negative numbers and blue represent negative correlation); asterisks represent significance (** stands for *p*-value less than 0.01).

**Figure 2 plants-12-03719-f002:**
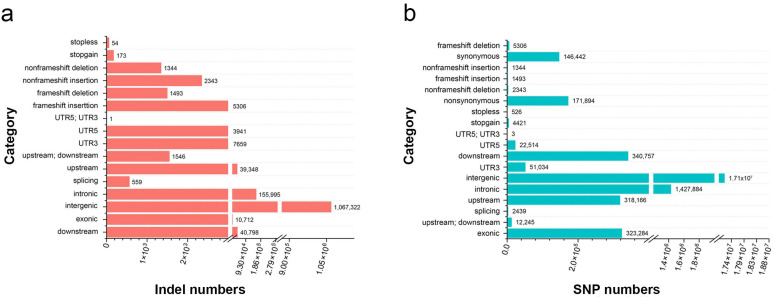
The statistics of indels and SNPs. (**a**) Statistical analysis of indels in the tea plant genome. (**b**) Statistical analysis of SNPs in the tea plant genome.

**Figure 3 plants-12-03719-f003:**
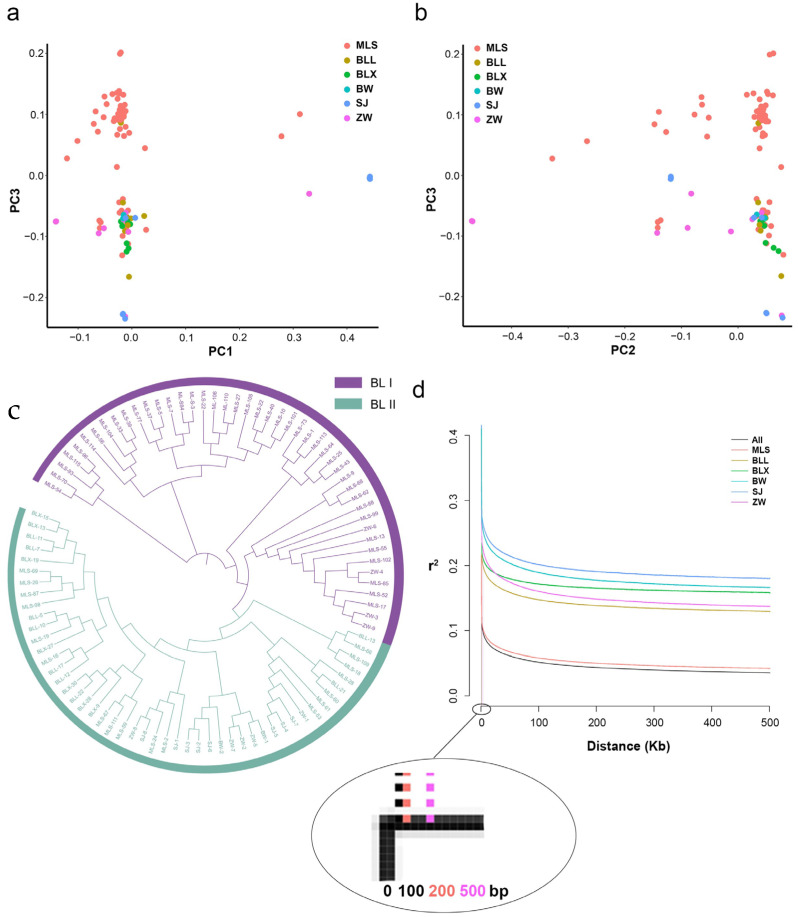
Genetic structure of 95 Biluochun tea plant varieties. (**a**) PCA plot presents the genetic relationship in tea plant varieties with PC1 and PC3. (**b**) PCA plot presents the genetic relationship in tea plant varieties with PC2 and PC3. Distribution of Biluochun tea plant varieties: 9 BLL, 7 BLX, 2 BW, 9 ZW, 8 SJ, and 60 MLS. (**c**) NJ tree of 95 ancient tea plants inferred from filtered SNPs. NJ tree shows the phylogenetic relationship among individuals. Purple members comprise BL I and light green members comprise BL II. BL I contains 46 tea plant individuals and BL II contains 49 tea plant individuals. (**d**) Genome-wide LD decay is estimated from all populations and six subpopulations. The *x*-axis represents the physical distance and the *y*-axis represents the average pairwise correlation coefficient (r^2^) of SNPs. The black, red, brown, green, cyan, blue, and purple represent All, MLS, BLL, BLX, BW, SJ, and ZW, respectively.

**Figure 4 plants-12-03719-f004:**
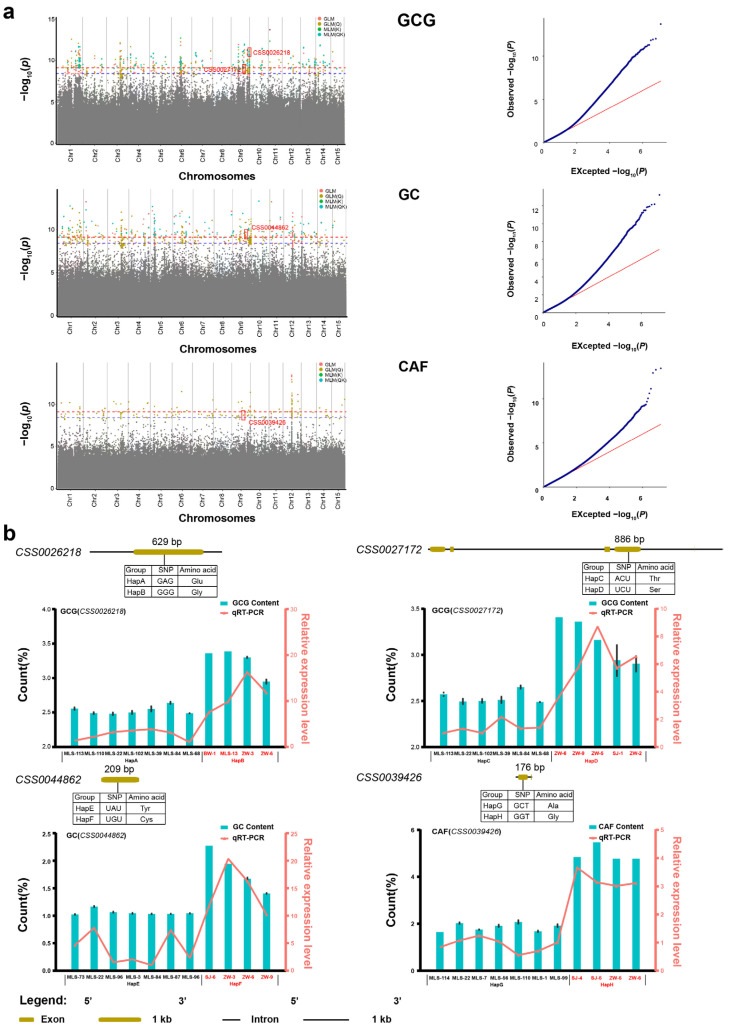
GWAS results about three metabolites, the distribution, structure, and expression of genes associated with the three metabolites in Biluochun tea plant varieties. (**a**) Manhattan and quantile–quantile plots. The blue and red dotted lines show significant associations between SNPs and metabolites value with threshold levels of *p*-value < 3.89 × 10^−9^ and *p*-value < 7.78 × 10^−10^, Red boxes indicate candidate SNPs and genes in manhattan plots. Red, yellow, green, and cyan dots represent GLM, GLM(Q), MLM(K) and MLM(QK). For quantile-quantile plots, the horizontal axis shows −log_10_ transformed expected *p* values, and the vertical axis indicates −log_10_ transformed observed *p* values. The red line represents the predicted value and the blue dot represents the observed value, which can show the difference between the predicted value and the observed value. (**b**) Candidate genes mutation information, the content of three metabolites, and the expression level of metabolite-related candidate genes in different haplotypes. Base and amino acid mutations and the structure of genes related to quality-related metabolites were plotted. The cyan bars represented the distribution of metabolites content and red lines indicated expression levels of metabolite-related candidate genes in different haplotypes.

**Table 1 plants-12-03719-t001:** Detailed information about the four genes associated with GCG, GC, and CAF were identified by GWAS.

Tarit	Gene Locus	Position	*p*-Value	Exon	SNP	Amino Acid	Gene Annotation
GCG	*CSS0026218*	164,656,928	6.02 × 10^−11^	1	GAG-GGG	Glu-Gly	LysM-containing receptor-like kinase
	*CSS0027172*	140,146,183	1.75 × 10^−9^	1	ACU-UCU	Thr-Ser	tubby-like protein transcription
GC	*CSS0044862*	144,566,106	4.57 × 10^−10^	1	UAU-UGU	Tyr-Cys	plant protein (DUF247)
CAF	*CSS0039426*	141,418,417	2.38 × 10^−9^	1	GCT-GGT	Ala-Gly	CREB-regulated transcription coactivator 2

## Data Availability

All the data generated or analyzed in this study are included in this published article and its Appendix A.

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
