# Peer review of "Genome-Wide Association Studies of Biluochun Tea Plant Populations in Dongting Mountain and Comprehensive Identification of Candidate Genes Associated with Core Agronomic Traits by Four Analysis Models"

_plants, 2023, doi:10.3390/plants12213719_

Round 1
Reviewer 1 Report
Comments and Suggestions for Authors
1 1. The aim of the study is to reveal genetic diversity, and different population structure of a collection of 95 tea plants (Biluochun tea).
2. There are no investigations about Biluochun tea plant populations. The structure of Biluochun tea population, its phenotype diversity, and its metabolites associated to quality traits need to be elucidated. The study is aimed to provide more knowledge about these research topics.
3. The metabolite analysis shows high variability among the 95 tea varieties. Furthermore, they are characterized by a wide variety of traits and genetic diversity. Therefore, Biluochun tea plants can be considered a great resource to study the metabolic and molecular mechanism of tea. This represents an added value of the proposed research.
4. The work is well described and all sections discussed in detail.
5. The conclusions are consistent with the achieved results, and underline how the innovative adopted GWAS approach allowed the identification of SNPs associated with the quality-related metabolites, and of the related candidate genes.
6. The references, the figures and tables are appropriate.
7. The work is very interesting and deserves to be published. However, some small modifications are suggested:
· In the Introduction, the primary and secondary metabolites of tea mentioned in the next sections could be better illustrated (for instance catechins)
· Lines 523-530. The abbreviation used in the manuscript could be showed into a table, thus increasing readability. Furthermore, other abbreviations could be added (for instance HPLC, PCC, qRT-PCR)
· Lines 531-561. Supplementary materials description could be shorter, or simply indicate where they can be found.
Comments on the Quality of English LanguageMinor editing of English language required
Author Response
|
Comments 1: The aim of the study is to reveal genetic diversity, and different population structure of a collection of 95 tea plants (Biluochun tea).
|
|
Response 1: Thank you for your comments and recognition.
|
|
Comments 2: There are no investigations about Biluochun tea plant populations. The structure of Biluochun tea population, its phenotype diversity, and its metabolites associated to quality traits need to be elucidated. The study is aimed to provide more knowledge about these research topics.
Response 2: Thank you for your comments and recognition.
|
|
Comments 3: The metabolite analysis shows high variability among the 95 tea varieties. Furthermore, they are characterized by a wide variety of traits and genetic diversity. Therefore, Biluochun tea plants can be considered a great resource to study the metabolic and molecular mechanism of tea. This represents an added value of the proposed research.
|
|
Response 3: Thank you for your comments and recognition.
|
|
Comments 4: The work is well described and all sections discussed in detail.
|
|
Response 4: Thank you for your comments and recognition.
Comments 5: The conclusions are consistent with the achieved results, and underline how the innovative adopted GWAS approach allowed the identification of SNPs associated with the quality-related metabolites, and of the related candidate genes.
Response 5: Thank you for your comments and recognition.
Comments 6: The references, the figures and tables are appropriate.
Response 6: Thank you for your comments and recognition.
Comments 7: In the Introduction, the primary and secondary metabolites of tea mentioned in the next sections could be better illustrated (for instance catechins).
Response 7: Thank you for pointing this out. We agree with this comment. Therefore, we have described the metabolic components of tea in the introduction. Please check the detailed information in lines 37-53. We have marked the revisions in red.
Comments 8: Lines 523-530. The abbreviation used in the manuscript could be showed into a table, thus increasing readability. Furthermore, other abbreviations could be added (for instance HPLC, PCC, qRT-PCR).
Response 8: Thank you for your suggestion. We agree with this comment. Therefore, we have rearranged the abbreviation used in the manuscript into a table. Please check the detailed information in the supplementary materials Table S6. We have marked the revisions (lines 564-565) in red.
Comments 9: Lines 531-561. Supplementary materials description could be shorter, or simply indicate where they can be found.
Response 9: Thank you for pointing this out. We agree with this comment. Therefore, we have simplified the supplementary materials description. Please check the detailed information in lines 553-565. We have marked the revisions in red.
|

Reviewer 2 Report
Comments and Suggestions for Authors
1. Lines 71 and 72. Full name of those abbreviations should be given.
2. Lines 79-81. The Shannon-Weaver index measures the metabolite diversity among tea varieties. I do not know how authors concluded high metabolite contents based on those values.
3. Lines 96 and 97, and figure S2. Please make sure the color code matching the one in figure 1.
4. Lines 99 and 100. More explanation is needed. What are the factors affecting tea quality?
5. Figure 1. The resolution of all figures need to be improved. Please check lines 91 and 94, figure 1 and figure S2, and make sure they match.
6. What does resequencing mean? Why resequencing matters should be given in the introduction.
7. Lines 148 and 149. It seems to be three groups in figure 3a.
8. Lines 170-172. This cannot be seen in figure 3d.
Author Response
|
Response to Reviewer 2 Comments
|
||
|
1. Summary |
|
|
|
Thank you very much for taking the time to review this manuscript. Please find the detailed responses below and the corresponding revisions/corrections highlighted/in track changes in the re-submitted files.
|
||
|
2. Questions for General Evaluation |
Reviewer’s Evaluation |
Response and Revisions |
|
Is the work a significant contribution to the field? |
|
|
|
Is the work well organized and comprehensively described? |
|
|
|
Is the work scientifically sound and not misleading? |
|
|
|
Are there appropriate and adequate references to related and previous work? |
|
|
|
Is the English used correct and readable? |
|
|
|
3. Point-by-point response to Comments and Suggestions for Authors |
|
|
|
Comments 1: Lines 71 and 72. Full name of those abbreviations should be given. Response 1: Thank you for pointing this out. We agree with this comment. Therefore, we have added all the full name. Please check the detailed information in lines 93-97. We have marked the revisions in red.
Comments 2: Lines 79-81. The Shannon-Weaver index measures the metabolite diversity among tea varieties. I do not know how authors concluded high metabolite contents based on those values. Response 2: Thank you for your question. We agree with this comment. The results of the contents of tea can indicate the level of content. However, the Shannon-Weaver index could show the diversity and abundance of tea metabolites. Therefore, we have modified this content. Please check the detailed information in lines 104-106. We have marked the revisions in red.
Comments 3: Lines 96 and 97, and figure S2. Please make sure the color code matching the one in figure 1. Response 3: Thank you for pointing this out. We agree with this comment. The color codes of Figure 1 and Figure S2 are slightly different, and we have unified the colors of the two figures. Please check the detailed information in the figure 1 and S2. We have marked the revisions in red.
Comments 4: Lines 99 and 100. More explanation is needed. What are the factors affecting tea quality? Response 4: Thank you for pointing this out. We agree with this comment. Therefore, we have explained several major metabolites that influence the formation of tea flavor and combined the results for further analysis. Please check the detailed information in lines 122-129. We have marked the revisions in red.
Comments 5: Figure 1. The resolution of all figures need to be improved. Please check lines 91 and 94, figure 1 and figure S2, and make sure they match. Response 4: Thank you for pointing this out. We agree with this comment. Therefore, we have improved the resolution of all figures and make sure they match. Please check the detailed information about all figures. We have marked the revisions in red.
Comments 6: What does resequencing mean? Why resequencing matters should be given in the introduction. Response 6: Thank you for pointing this out. We agree with this comment. Therefore, we have described the meaning of resequencing in the introduction. Please check the detailed information in lines 54-67. We have marked the revisions in red.
Comments 7: Lines 148 and 149. It seems to be three groups in figure 3a. Response 7: Thank you for pointing this out. The vast majority of tea plant individuals in figure 3a are divided into two groups, and several individuals on the right can be defined as outlier samples. We can refer to Guizhou tea plant resequencing, doi: 10.1038/s41438-021-00617-9.
Comments 8: Lines 170-172. This cannot be seen in figure 3d. Response 8: Thank you for pointing this out. We agree with this comment. We have marked LD decay distance in Figure 3d. Please check the detailed information in figure 3d We have marked the revisions in red.
|
||